# Wnt/β-Catenin-Signaling Modulates Megakaryopoiesis at the Megakaryocyte-Erythrocyte Progenitor Stage in the Hematopoietic System

**DOI:** 10.3390/cells12232765

**Published:** 2023-12-04

**Authors:** Burak H. Yalcin, Jadranka Macas, Eliza Wiercinska, Patrick N. Harter, Malak Fawaz, Tessa Schmachtel, Ilaria Ghiro, Ewa Bieniek, Djuro Kosanovic, Sonja Thom, Marcus Fruttiger, Makoto M. Taketo, Ralph T. Schermuly, Michael A. Rieger, Karl H. Plate, Halvard Bonig, Stefan Liebner

**Affiliations:** 1Institute of Neurology (Edinger Institute), University Hospital Frankfurt, Goethe University, 60590 Frankfurt am Main, Germanymacas@em.uni-frankfurt.de (J.M.); ilariaghiro@icloud.com (I.G.); plate@em.uni-frankfurt.de (K.H.P.); 2Institute for Transfusion Medicine and Immunohaematology, and DRK-Blutspendedienst BaWüHe, Goethe University Frankfurt, 60528 Frankfurt am Main, Germany; 3Department of Medicine, Hematology/Oncology, University Hospital Frankfurt, Goethe University, 60590 Frankfurt am Main, Germanym.rieger@em.uni-frankfurt.de (M.A.R.); 4German Center for Lung Research (DZL), Department of Internal Medicine, Excellence Cluster Cardio-Pulmonary Institute (CPI), Justus Liebig University of Giessen, Aulweg 130, 35392 Giessen, Germany; ewa.bieniek@innere.med.uni-giessen.de (E.B.); djurokos13@gmail.com (D.K.);; 5UCL Institute of Ophthalmology, London EC1V 9EL, UK; m.fruttiger@ucl.ac.uk; 6Kyoto University Hospital-iACT Graduate School of Medicine, Kyoto University, Kyoto 06-8501, Japan; 7German Cancer Consortium (DKTK) at the German Cancer Research Center, 69120 Heidelberg, Germany; 8Frankfurt Cancer Institute (FCI), 60596 Frankfurt am Main, Germany; 9Excellence Cluster Cardio-Pulmonary Institute (CPI), Partner Site Frankfurt, 60590 Frankfurt am Main, Germany; 10German Center for Cardiovascular Research (DZHK), Partner Site Frankfurt/Mainz, 60590 Frankfurt am Main, Germany; 11Department of Medicine/Division of Hematology, University of Washington, Seattle, WA 98195, USA

**Keywords:** megakaryocytes, erythrocytes, β-catenin, Wnt signaling, cell-fate decision, maegakaryopoiesis

## Abstract

The bone marrow (BM) hematopoietic system (HS) gives rise to blood cells originating from hematopoietic stem cells (HSCs), including megakaryocytes (MKs) and red blood cells (erythrocytes; RBCs). Many steps of the cell-fate decision remain to be elucidated, being important for cancer treatment. To explore the role of Wnt/β-catenin for MK and RBC differentiation, we activated β-catenin signaling in platelet-derived growth factor b (Pdgfb)-expressing cells of the HS using a Cre-lox approach (Ctnnb1^BM-GOF^). FACS analysis revealed that Pdgfb is mainly expressed by megakaryocytic progenitors (MKPs), MKs and platelets. Recombination resulted in a lethal phenotype in mutants (Ctnnb1^BM-GOFwt/fl^, Ctnnb1^BM-GOFfl/fl^) 3 weeks after tamoxifen injection, showing an increase in MKs in the BM and spleen, but no pronounced anemia despite reduced erythrocyte counts. BM transplantation (BMT) of Ctnnb1^BM-GOF^ BM into lethally irradiated wildtype recipients (BMT-Ctnnb1^BM-GOF^) confirmed the megakaryocytic, but not the lethal phenotype. CFU-MK assays in vitro with BM cells of Ctnnb1^BM-GOF^ mice supported MK skewing at the expense of erythroid colonies. Molecularly, the runt-related transcription factor 1 (RUNX1) mRNA, known to suppress erythropoiesis, was upregulated in Ctnnb1^BM-GOF^ BM cells. In conclusion, β-catenin activation plays a key role in cell-fate decision favoring MK development at the expense of erythroid production.

## 1. Introduction

Hematopoiesis is the production of mature blood cells from hematopoietic stem cells (HSCs). As cells proliferate and mature, they reach branching points restricting their differentiation potential. At a relatively late step of fate decision, common myeloid progenitors (CMPs) have the ability to mature via megakaryocyte-erythroid progenitors (MEPs) into megakaryocytic or erythroid/red blood cell progenitors (MKPs, RBCPs), finally maturing to megakaryocytes (MKs) or erythrocytes/red blood cells (RBCs), respectively [1,2,3]. At this stage, an intricate interplay of positive and negative transcription factors initiates the definitive determination towards the erythroid vs. megakaryocytic lineage [4,5]. While significant insight into lineage determination has been gleaned, many details remain elusive. Especially with respect to the Wnt/β-catenin system, seemingly contradictory data have been reported, resulting in confusion as to the exact contribution of this conserved signaling pathway to mature hematopoiesis. However, understanding molecular mechanisms of cellular differentiation is key in defining novel targets for the treatment of neoplasms in the hematopoietic system. Several forms of leukemic neoplasms show a high association with the Wnt/β-catenin pathway. For instance, primary cells from patients with chronic myelogenous leukemia (CML) were shown to exhibit significantly elevated levels of Wnt-signaling activity [6]. Moreover, CML patients display a mis-splicing of GSK-3β RNA, leading to the deletion of exons 8 and 9, and resulting in a truncated form of GSK-3β that lacks the axin binding domain, thereby preventing β-catenin phosphorylation and proteasomal degradation, hence, excessive and protracted Wnt pathway activation [7]. Also, acute myeloid leukemia (AML) was observed to depend on Wnt/β-catenin, as the genetic deletion of β-catenin reduced leukemia incidence and decreased stem cell self-renewal in AML mouse models [8]. Inversely, the dominant activation of β-catenin specifically in osteoblasts resulted in leukemogenesis in mice, suggesting a crucial function of the canonical Wnt pathway in the pathophysiology of the bone marrow (BM). This observation is in line with the well-known function of the canonical Wnt pathway in the determination of cell fate, proliferation, polarity and cell death. Herein, Geduk and colleagues demonstrated that MKs and endothelial cells display elevated β-catenin immunoreactivity in BM samples from primary myelofibrosis (PMF), polycythemia vera (PV) and essential thrombocythemia (ET) patients [9]. Interestingly, recent data by Heil et al. propose that Wnt/β-catenin signaling in BM sinusoidal endothelial cells (BM-SEC) controls terminal erythroid and reticulocyte maturation [10]. Although the authors observed an increase in MEPs, they neither observed increased MK formation nor increased platelet counts.

Furthermore, it remains to be clarified in which cell types of the BM the Wnt/β-catenin pathway becomes activated during normal hematopoiesis and in neoplastic transformations such as in leukemic processes. Depending on the use of cell-specific Cre-driver lines in mouse models, or in cellular systems, conflicting data regarding the outcome of Wnt/β-catenin activation have challenged the understanding of its function in the hematopoietic system [11,12]. In the present manuscript, we studied the effect of constitutive Wnt activation exclusively in cells expressing PDGF, i.e., for all intents and purposes exclusively in platelet lineage cells on mature hematopoiesis. Hematopoiesis was megakaryocyte skewed at the expense of erythropoiesis. We demonstrate that activated Wnt-signaling acts as a modulator of cell fate by reinforcing Runx1 and, to some degree, Fli1 transcription factors which are described as key regulators in the megakaryocytic differentiation program. Dominant activation of β-catenin signaling in CFU assays in vitro corroborated this finding. These results give new insights into the understanding of the development of myeloproliferative neoplasms, such as polycythemia vera and essential thrombocytopenia, which are among other attributes, characterized by increased MK and platelet levels in the hematopoietic system.

## 2. Materials and Methods

### 2.1. Animals and Transgene Induction

Animals were housed under standard conditions and fed ad libitum. All experimental protocols, handling and use of mice were approved by the Regierungspräsidium Darmstadt, Germany (V54-19c20/15-FK/1069). The transgenic mouse strains used in the project include PDGFb-iCreERT2 [13], Cdh5(PAC)-CreERT2 [14], Ctnnb1^wt/lox(ex3)^ and Ctnnb1^lox(ex3)/lox(ex3)^ [15], as well as wild type (WT) C57BL/6N Ly5.2 or Ly5.1.

To achieve the constitutive activation of β-catenin in the BM, Ctnnb1^lox(ex3)^ mice were crossed with the PDGFb-iCreERT2 mouse line [13], resulting in the Ctnnb1^BM-GOFwt/fl^ or ^fl/fl^ double transgenic mouse line. Additionally, to achieve endothelial-specific β-catenin activation, Ctnnb1^lox(ex3)/lox(ex3)^ [15] mice were crossed with the Cdh5(PAC)-CreERT2 [14] line, resulting in the Ctnnb1^EC-GOFwt/fl^ or ^fl/fl^ double transgenic mice.

The PDGFb-iCreERT2 transgene contains an internal ribosome entry site (IRES), leading to the expression of enhanced green fluorescent protein (EGFP) in the open reading frame driven by the PDGFb promoter [13].

For both Cre lines used, the Cre allele was inherited heterozygously in double-transgenic mice. Littermates negatively genotyped for Cre expression served as controls (CTL).

Tamoxifen Free Base (MP Biomedicals, Irvine, CA, USA #156738) was dissolved in corn oil for a final concentration of 5 mg/mL; 100 μL/day/mouse (25 mg/kg) were intraperitoneally (i.p.) injected on 5 subsequent days, as shown in Figure 1 [13,16]. Tamoxifen-mediated deletion of exon3 from the Ctnnb1 gene results in the translation of a dominant-active β-catenin protein, lacking the N-terminal domain required for protein degradation [15].

### 2.2. Organ Dissection, Cryopreservation and Paraffin Embedding

Mice were sacrificed by cervical dislocation upon a brief isoflurane anesthesia.

Organs and tissues were removed and natively embedded in Tissue-Tek^®^ O.C.T. compound, or tissues were fixed in 4% PFA overnight (o/n) and processed for either Tissue-Tek^®^ or paraffin embedding, as previously described [17,18]. Tissue-Tek^®^-embedded samples were stored at 80 °C.

### 2.3. Histology and Immunofluorescence

#### Preparation of Cryosections

Tissue-Tek^®^-embedded samples were sectioned (12 μm) using a cryostat (Microm HM550 OMVP) and collected on microscopic glass slides (SUPERFROST^®^ PLUS).

### 2.4. Bone Decalcification

Bones (femur) were postfixed in 4% PFA o/n at 4 °C. The decalcification process was performed in an EDTA solution (10% EDTA/10 mM Tris-HCL, pH 6.8–7.0) at 37 °C. Finally, the decalcified bones were washed for 1 week in PBS at 4 °C and embedded in Tissue-Tek^®^ O.C.T. for 12 µm thick cryosections. Sections were washed in PBS and processed for immunofluorescent staining as described below.

### 2.5. Complete Blood Cell (CBC) Analysis

Peripheral blood samples of animals (min. 50 μL) were analyzed using HemaVet 950FS Auto Blood Analyzer (CBC and four-way differential). Raw data from individual mice were combined in GraphPad Prism 6.0 in which the mean, standard deviation (SD) and statistics were calculated and plotted as graphs.

### 2.6. Immunofluorescence Staining (IFS)

Longitudinal femur sections (12 μm) were fixed in 4% paraformaldehyde (PFA) for 10 min, depending on the requirements of the antibodies used (see Table 1), and subsequently washed in PBS, followed by 1 h incubation in permeabilization–saturation buffer (5% goat serum in PBS with 0.1% Triton). Primary antibodies (for detail see Table 1) were incubated for 1–2 h at RT or overnight at 4 °C in 0.5% BSA in PBS 0.1% TritonX-100, and secondary antibodies for 2 h at RT. Following nuclear staining with 4′,6-Diamidin-2-phenylindol (DAPI), slides were mounted with Aqua-Poly/Mount (Polysciences, Warrington, PA, USA #18606) or Fluoromount-G (#00-4958-02, Invitrogen, Darmstadt, Germany).

### 2.7. Immunohistochemistry Staining (IHC)

IHC was performed on the Leica BOND III automated staining system (Leica Mikrosysteme Vertrieb GmbH, Wetzlar, Germany) using standard protocols. Haematoxylin–Eosin (HE) staining, Pappenheim staining, Picro-Sirius red and Prussian blue staining were carried out according to standard procedures.

### 2.8. Image Acquisition and Processing

Light microscopic images were acquired using a Nikon 80i wide-field microscope. Immunofluorescence images were acquired using Nikon C1si or A1RHD25 Confocal Laser Scanning Microscope Systems. Images were processed with NIS Elements AR Microscope Imaging Software 5.21.01 (Nikon Instruments, Inc., Düsseldorf, Germany), Affinity Photo 1.1 and arranged with Affinity Designer 1.1 (Serif (Europe) Ltd., The Software Centre, 12 Wilford Ind Est, Nottingham, UK.).

### 2.9. Preparation of Blood and BM Samples for FACS Analysis

#### Detection of Blood Cell Types

Blood was drawn from the *Vena maxillaris* into EDTA-coated blood tubes. A fraction was used for CBC, the remainder was subjected to hypotonic lysis and, after washing, to antibody incubation for FACS analysis. After washing in FACS buffer (PBS + 5% fetal calf serum, FCS), primary antibodies (Table 1) were applied for 1 h. Subsequently, cells were washed 1x in FACS buffer and dissolved in PBS for FACS analysis (see Appendix A for gating strategy).

### 2.10. Preparation of BM Samples for FACS Analysis

Femurs were dissected out of adult mice. Remaining muscle tissue on the bones was removed and both bone ends were cut to flush the BM with FACS buffer. The cell suspension was filtered (100 μm cell strainer) and centrifuged for 5 min (1200 rpm, 4 °C). The pellet was re-suspended in 100 μL FACS buffer and incubated with antibodies for 1 h at 4 °C. After the incubation, samples were centrifuged (1200 rpm, 5 min, 4 °C), re-suspended in 500 μL PBS and analyzed using FACS Aria or FACS Canto II (BD Biosciences, Heidelber, Germany). A minimum of ~1 × 10^7^ events per sample were recorded.

### 2.11. Preparation of Spleen Samples for FACS Analysis

The spleen was dissected out and homogenized between two glass slides. Cell suspension was re-suspended in FACS buffer and transferred into FACS tubes. After centrifugation (1200 rpm, 4 °C), the pellet was re-suspended in 1 mL red blood cell lysis buffer (Sigma-Aldrich/Merck, Darmstadt, Germany; #11814389001) for 10 min at RT, centrifuged and washed with FACS buffer. The pellet was re-suspended and incubated with desired antibodies for 1 h on ice, washed and re-suspended in 1 mL PBS for FACS analysis.

### 2.12. Detection of GFP^+^MKPs/MKs

FACS analysis of the PDGFb construct expressing cells in the spleen was performed on the FACS Canto TM II using the following strategy (see also Figure 3): the first gate (A1) includes BM cells using forward scatter light (FSC) and sideward scatter light (SSC). BM cells in A1 were gated for the GFP reporter signal (A2) and plotted for CD41^+^ (A4) CD150^+^ (A5) MKPs. Remaining GFP^+^ cells were quantified by gate A3. Finally, MKPs (A5) were analyzed for granularity and cell volume by SSC and FSC (A6). The back-gating of MKPs for the GFP signal was achieved by plotting cells in A1 first for CD150 (A2) and then for CD41 (A4). Finally, CD41^+^/CD150^+^ cells were analyzed for the GFP signal (A5) (Appendix A).

MKPs in the spleen and BM were quantified on the FACS Canto TM II using the following gating strategy: spleen cells were detected (A1) and dead cells were dismissed using DAPI (A2). CD150+/CD41+ MKPs were gated (A3, A4) and plotted for granularity and cell volume in gate A5. Staining specificity was evaluated by the fluorescence minus one (FMO) controls and Cre-negative cells, respectively (Appendix A).

### 2.13. Detection of HSCs and Progenitor Cells in the BM

The gating strategy (see also Figure 4A) for HSCs and progenitor cells was performed on the FACS AriaTM: BM cells were gated for the FSC and SSC signal (A1). In the next step, lin-negative cells (A2) were gated and plotted for Sca-1 and c-Kit to detect the fractions KL (CMPs, GMPs, MEPs) (A3) and KSL (LT-HSCs, ST-HSCs, MPPs) (A4). The KL fraction was gated for the CD16/32 marker to distinguish between the CMPs, GMPs and MEPs (A5).

### 2.14. RNA Isolation and RT-PCR and qRT-PCR Analyses

The purification of RNA was performed by using the Rneasy Micro Kit from QIAGEN® according to the Quick-StartProtocol. For cDNA amplification, the cDNA Synthesis Kit (K1632) from ThermoFisher Scientific was used according to the manufacturer’s protocol. Specific primers are listed in Table 2. qRT-PCR was performed in a C1000™ Thermal cycler (CFX96™ Real-Time System). Reference genes were 18s and Rplp0, whereas RNA from an E14.5 mouse embryo served as positive control. RT-PCR products were loaded on 1.5% agarose gels and pictures were taken with a MultiImageTM Light Cabinet (FluorChem, Alpha Innotech/ProteinSimple, Silicon Valley, CA, USA).

Quantitative real-time RT-PCR (qRT-PCR) was performed using the Absolute qPCR SYBR Green Fluorescein Mix (AB-1219, ThermoFisher Scientific, Darmstadt, Germany) according to the manufacturer’s protocol in a CFX96 real-time PCR device (Biorad, Feldkirchen, Germany). Expression data were analyzed as a normalized expression to 18s RNA as a housekeeping gene and statistics were performed in GraphPad Prism 6.0 software. Primer sequences used for cDNA amplification by qRT-PCR are listed in Table 2.

### 2.15. Bone Marrow Transplantation (BMT)

Recipients from the Ly5.1 mouse line were lethally irradiated with 9.5 Gy in a single dose with a Cesium-137 source at a dose rate of 0.7 Gy/min (Biobeam 2000, Gamma Medical). Right after irradiation, BM cells (10^6^ cells) from non-tamoxifen-induced single transgenic animals labeled as BMT CTL (PDGFb-iCreERT2::Ctnnb1-Ex3^wt/wt^; *n* = 7) or heterozygous double transgenic donors labeled as BMT Ctnnb1^BM-GOFwt/fl^ (PDGFb-iCreERT2::Ctnnb1-Ex3^wt/fl^; *n* = 3), as well as homozygous double transgenic BMT Ctnnb1^BM-GOFfl/fl^ (PDGFb-iCreERT2::Ctnnb1-Ex3^fl/fl^; *n* = 3) were transplanted by i.v. injection into recipients. After sable engraftment, 12–16 weeks after transplantation, animals were treated with 25 mg/kg tamoxifen per day in two sessions starting with days 1–5, followed by days 29–33 to induce the BM specific Cre recombination (Figure 5A,B). The engraftment was analyzed in PB by FACS analysis: in the first step, PB cells (A1) were checked whether they were produced by the recipient (A2: CD45.1^+^) or transplanted (A3: CD45.2^+^) BM cells. Furthermore, lineage-specific engraftment was investigated by a myeloid (A6), B cell (A4) and T cell (A5) marker (Appendix A).

### 2.16. Detection of MKPs in BMT Animals

BM cells were prepared as described above and stained with markers listed in Table 1. To quantify specifically transplanted cells from donors, CD45.2^+^ (A2) events were selected out of BM cells (A1). In the next step, MKPs (A4 and A5) were counted out of CD45.2^+^ cells and measured for the PDGFb construct GFP reporter (A6) (Appendix A).

### 2.17. CFU—Assay Analysis

The following procedure was used to process one mouse with ~2 × 10^8^ cells per separation.

The tibia, femur and pelvic bones of each mouse were prepared and thoroughly cleaned from connective tissue. BM was flushed with 3 mL PBS, and the suspension was carefully loaded on top of 3 mL Histopaque 1083 Ficoll-gradient and centrifuged for 30 min at RT at 400 g. The interphase containing mononucleated cells was washed 3x in PBS, counted and seeded at 1 × 10^4^ cells/1.5 mL in MethoCult™ (STEMCELL #03434), including 25 μM 4-hydroxy tamoxifen (4-OHT) and 100 ng/mL TPO. Scoring of colonies was performed after 10 days of cultivation according to their size and specific cell shapes.

In parallel, mouse brain microvascular endothelial cells (MBMECs) were isolated from the same mice, cultivated as previously carried out [17], treated with 25 μM 4-OHT and harvested after 7 days for gDNA and mRNA analysis, serving as a positive control for Pdgfb-iCreERT2-mediated recombination of the Ctnnb1^lox(ex3)^ locus, as well as for Wnt/β-catenin pathway activation. Cre-mediated recombination was confirmed by PCR on genomic DNA (gDNA) as previously described [15], using the following primers: β-catGF2 (Ex3_wt/rec02s) GGTACCTGAAGCTCAGCGCACAGC; β-catAS5 (Ex3_wt/rec02as) ACGTGTGGCAAGTTCCGCGTCATCC. mRNA served to monitor the Axin2 Wnt/β-catenin target gene induction, using the following primers: Axin2s GCCGACCTCAAGTGCAAACTC; Axin2as GGCTGGTGCAAAGACATAGCC-AS (Appendix A).

### 2.18. Colony Identification and Scoring

The analysis was performed as described in the technical manual provided by STEMCELL™ Technologies to identify the colony types:

CFU-E: colony-forming unit–erythroid; CFU-GM: colony-forming unit–granulocyte, macrophage; CFU-GEMM: colony-forming unit–granulocyte, erythroid, macrophage, megakaryocyte; CFU-MK: colony-forming unit–megakaryocyte. At least 3 or more megakaryocytes are necessary to consider it a colony. The cells develop MK-specific shape with granulocytic cell bodies and matured proplatelets.

### 2.19. Statistics

Statistics were performed in GraphPad Prism 6.0 software, using a t-test with Welch’s correction for column comparison. Graphs were also generated in Prism 6.0 and exported as TIF files (300 dpi). Graph are shown with standard deviation (SD), except for stacked graphs, for which for better clarity the standard error of the mean (SEM), is shown.

### 2.20. Graphics and Artwork

Figures and drawings were assembled in Affinity Designer (Serif Europe Ltd., West Bridgford, UK).

## 3. Results

### 3.1. PDGFb-iCreERT2-Mediated Wnt/β-Catenin Pathway Activation in Mice Led to a Lethal Phenotype, Erythrocyte Reduction and Increased Megakaryopoiesis

To achieve constitutive β-catenin activation in PDGFb-expressing cells in the bone marrow, PDGFb-iCreERT2 mice were crossed to the Ctnnb1^lox(ex3/ex3)^ mouse strain, resulting in the tamoxifen (TAM)-inducible heterozygous and homozygous Ctnnb1^BM-GOFwt/fl^ or Ctnnb1^BM-GOFfl/fl^ strain (Figure 1A). Adult mice received intraperitoneal (i.p.) tamoxifen injections on 5 consecutive days and were analyzed 3 weeks after the last injection or upon the formation of symptoms that required sacrificing the mice (Figure 1B). Consistently, TAM induction of constitutive β-catenin activity led to a lethal phenotype in mutant Ctnnb1^BM-GOFwt/fl^ and Ctnnb1^BM-GOFfl/fl^ animals about 3 weeks after the last TAM injection. Symptoms were impaired locomotion and dyspnea. Additionally, GOF animals showed impaired hair growth with flaking and irritated skin (Figure 1C), which was characterized as hyperkeratosis on skin sections (Appendix A). Along the body axis, the major anatomical sites for hyperkeratosis were dorso-lateral and dorso-caudal (Figure 1C). Analysis of HE-stained skin samples revealed a high level of keratinization in the epidermal layer with dystrophic hair follicles, resulting in impaired hair growth in BM-GOF mice (Appendix A). The epidermal layer and single cells in the dermal parenchyma showed augmented Ki67 staining, pointing to increased proliferation (Appendix A, arrows). Furthermore, lung sections were examined to clarify if the breathing phenotype in mutant animals might be related to pathological changes. HE staining served to visualize gross morphological abnormalities (Appendix A), Picro-sirius staining for connective tissue (Appendix A) and CD3 T cell staining for immuno-response (Appendix A), revealing no noticeable lung pathology in GOF animals.

Given that skin symptoms likely did not account for the lethal phenotype of the mice, we focused on the hematopoietic system as another known site of PDGFb expression [13].

To trace alterations in the hematopoietic system, TAM was injected into 12–16-week-old animals, and the spleen, BM and peripheral blood (PB) were harvested from Ctnnb1^BM-GOFwt/fl^ and Ctnnb1^BM-GOFfl/fl^ mice 3 weeks after induction. Since the spleen is known to be a hematopoietic organ with functions such as terminal differentiation of erythrocytes and sequestration of aged erythrocytes, it can serve as an indicator of malfunctions in blood development. Indeed, mutant mice developed splenomegaly, which was confirmed by visual examination and by a spleen/body weight ratio (Figure 1D,E). Upon diagnosing splenomegaly in Ctnnb1^BM-GOFfl/fl^ mutants, PB samples were analyzed by a complete blood cell (CBC) assay. Ctnnb1^BM-CTL^ and Ctnnb1^BM-GOF^ animals showed a normal concentration of thrombocytes (normal range: 592–2972 K/μL), in addition to a trend of increased counts in Ctnnb1^BM-GOFwt/fl^ mice (Figure 1F). Instead, the red blood cell count was reduced in Ctnnb1^BM-^GOF mice, suggesting skewing towards a megakaryocytic fate at the expense of erythropoiesis, almost completely compensated at the terminal differentiation stage (Figure 1G). However, more robust values such as hemoglobin (Hb) and hematocrit (HCT) did not show significant alterations, i.e., the mice did not develop outright anemia (Appendix A). In the next step, spleen sections were prepared for a Pappenheim staining to evaluate the presence of hemosiderin and hemosiderophages that would be indicative of spleen hemorrhage. Quantification of the hemosiderin per high-power field (HPF) revealed a two times higher amount of hemosiderin coverage in CTL than Ctnnb1^BM-GOFwt/fl^ and Ctnnb1^BM-GOFfl/fl^ mice, which were at comparable levels (Figure 1H,I).

In addition to the CBC assays, FACS analysis was performed on the same PB samples to specifically analyze myeloid and lymphoid cells (Figure 1J). In particular, the quantity of granulocytic cells was significantly increased in Ctnnb1^BM-GOF^ mice. Consequently, a shift in cell counts from the lymphoid towards the myeloid lineage was observed (Figure 1J). The level of B cells in the lymphoid compartment remained unaffected (Figure 1J). Homogenized spleen samples from the same animals corroborated the increase in myeloid cells (monocytes) in Ctnnb1^BM-GOF^ mice.

Based on the lower erythrocyte counts and in Ctnnb1^BM-GOF^ mice, we aimed to analyze the MKs as they represent the second lineage that is derived from a common megakaryocyte–erythroid progenitor (MEP). Initially, MKs were analyzed by either immunofluorescent staining (IFS) or FACS. The number of CD41^+^ cells in spleen cryosections was significantly higher in Ctnnb1^BM-GOFwt/fl^ and Ctnnb1^BM-GOFfl/fl^ compared to CTL animals when quantified by microscopic counting of MKs/HPF (Figure 2A,B). This finding was confirmed by FACS analysis of spleen samples from Ctnnb1^BM-GOFwt/fl^ and Ctnnb1^BM-GOFfl/fl^ (Figure 2C), as well as in the BM from Ctnnb1^BM-GOFfl/fl^ mice (Figure 2D), speaking in favor of a lineage skewing towards MKs.

To understand whether the increase in MKs would coincide with the promoter activation and concomitant Cre recombinase expression, we investigated the GFP reporter in these MKs which is expressed along by an IRES element. As originally suggested by Claxton et al., analysis of GFP expression by FACS in whole spleen samples gated for the transgene (PDGFb-iCreERT2) GFP reporter confirmed that most GFP^+^ cells had a megakaryocytic signature (CD41^+^, Figure 3A).

Interestingly, it turned out that the same cell population also exhibited a progenitor cell phenotype (CD41^+^/CD150^+^, Figure 3A). In turn, the back-gating of CD41^+^/CD150^+^ cells for GFP showed that almost all (≥99%) of them were GFP^+^ (Figure 3B, Appendix A). These CD41^+^/CD150^+^ cells showed high granularity and appeared to be relatively large, which was in line with megakaryocytic properties (Figure 3B). Thus, these data support the interpretation that the PDGFb transgene (GFP reporter) is mainly active in MK progenitor cells (MKPs), which are described as CD41^+^/CD150^+^ [19]. Additionally, IFS on BM cryosections was performed confirming the GFP reporter expression in CD41^+^ cells (Figure 3C). Interestingly, there was a group of not further characterized cells featuring CD41^-^/GFP^+^ characteristics (Figure 3C). The CD41^-^/GFP^+^ cell population was also identified in FACS analysis of the BM from Ctnnb1^BM-GOF^ mice (Figure 3A, gate #A3). Cre-negative Ctnnb1^BM-CTL^ mice did not show any GFP^+^ cells, nor did the respective antibody controls for CD41 and CD150 (Figure 3D–F).

Given the fact that the GFP reporter within the PDGFb transgenic construct is present in MKPs, further progenitors such as GMPs, CMPs and MEPs as well as HSCs were analyzed to investigate cells in the higher order of the hematopoietic hierarchy. For this, a well-established protocol for FACS analysis was used, which allowed to distinguish HSCs and downstream progenitors (Figure 4A). Comparing Ctnnb1^BM-CTL^ and Ctnnb1^BM-GOF^ BM samples showed no differences for HSCs, CMPs and GMPs regarding GFP expression (Figure 4A–D). However, the MEP pool was decreased in Ctnnb1^BM-GOF^ animals (Figure 4E).

### 3.2. Bone Marrow Transplantation (BMT) from Ctnnb1^BM-GOF^ Mice into Lethally Irradiated Controls Corroborated Increased Megakaryopoiesis in Mutant BM Cells

To achieve BM-specific recombination, BM cells from double transgenic donors (CD45.2^+^, Ctnnb1^BM-GOF^) and single transgenic donors (CD45.2^+^, CTL) were transplanted into lethally irradiated Ly5.1 (CD45.1^+^) recipient mice (BMT_Ctnnb1^BM-GOFwt/fl^, BMT_Ctnnb1^BM-GOFfl/fl^, BMT_CTL) (Figure 5A). The successful engraftment of the newly transplanted BM was analyzed by FACS for recipient (CD45.1^+^) and donor (CD45.2^+^) leukocytes in PB samples taken 12 weeks after BMT (see Appendix A for gating strategy). Gating all PB cells revealed that the majority of the PB cells originated from CD45.2^+^ donors in both the BMT_CTL and BMT_Ctnnb1^BM-GOFwt/fl^ (Appendix A). Most lymphoid cells (T cells and B cells) and myeloid cells originated from CD45.2^+^ donors, showing that the recipient BM was successfully engrafted (Appendix A).

After confirming the successful BMT, the recipient mice were treated twice with tamoxifen and sacrificed 3 weeks after the last injection (Figure 5B). Interestingly, mutant recipient animals (BMT_Ctnnb1^BM-GOF^) did not develop breathing difficulties, splenomegaly or a lethal phenotype, indicating that these symptoms did not develop due to alterations in the hematopoietic system initially seen in Ctnnb1^BM-GOF^ animals (Appendix A).

Complete blood count (CBC) assays showed no changes concerning the presence of thrombocytes and erythrocytes between BMT_Ctnnb1BM-GOF and BMT_Ctnnb1BM-CTL mice (Appendix A). However, it is important to mention that the erythroid level in BMT_Ctnnb1BM-GOFwt/fl animals, while not overtly anemic, mean RBC counts were very mildly anemic (mean, 6.8 M/µL, normal range 6.36–9.42 M/µL), whereas control mice were well within the range of normal (8.2 M/μL). Along this line, white blood cells (WBCs) in BMT_Ctnnb1^BM-GOFwt/fl^ tended to decrease while the recipients transplanted with homozygous BM (BMT_Ctnnb1^BM-GOFfl/fl^) remained at the same level as the CTLs (Appendix A).

The initial Ctnnb1^BM-GOF^ animals showed a significant decrease in the MEP pool compared to CTLs (Figure 4E). To prove whether the BMT recipients develop changes in the HSC and progenitor cells as well, FACS analysis by a well-established gating strategy (Figure 4A) revealed comparable levels in HSCs and progenitors indicating no significant differences with respect to the more primitive subsets between BMT_Ctnnb1^BM-CTL^ and BMT_Ctnnb1^BM-GOF^ (Appendix A).

Importantly, however, BMT and BM-specific activation of Wnt/β-catenin signaling in MKP cells led to increased megakaryopoiesis (Figure 5C). This was evaluated by gating BM of the recipient animals for donor cells (CD45.2^+^) and, additionally, MKPs were counted within these cells to guarantee donor-specificity (Appendix A). In the spleen, MKPs were significantly increased in BMT_Ctnnb1^BM-GOFwt/fl^ mice, while MKPs in the BM only in trend increased (Figure 5C,D).

To confirm whether the PDGFb-iCreERT2 transgene was active in the donor MKPs the gating strategy was also involved in the PDGFb-iCreERT2 transgene-inherited GFP reporter. Plotting all counted MKPs for the GFP signal showed that in the BMT_Ctnnb1^BM-GOF^ group, all MKPs expressed the PDGFb transgene, whereas MKPs from the BMT_Ctnnb1^BM-CTL^ group were negative, confirming the activity of the PDGFb construct in BMT (Appendix A).

To further support the specificity of the Wnt/β-catenin pathway in the bone marrow excluding other stromal cells, we have activated Cre to achieve β-catenin truncation under the highly specific vascular cadherin (VE-cadherin, CD144) promoter in endothelial cells (Appendix A). This did not result in significant changes in MkPs in the BM, nor in the spleen after 3 weeks of tamoxifen injection (Appendix A).

These data suggest that the transcriptional activity of β-catenin in MKPs regulates the differentiation and maturation of MKs in the BM of mice. However, whether this is a cell intrinsic effect or might be influenced by the other stromal cells in the BM such as blood vessels and others, was unclear.

### 3.3. Activating Wnt/β-Catenin Signaling in Ctnnb1^BM-GOF^ Cells In Vitro Leads to Enhanced Megakaryopoiesis and Decreased Erythropoiesis

To address the question of cell-autonomous effects, and if dominant Wnt/β-catenin activation during hematopoietic linage differentiation fosters MK formation at the expense of erythrocyte/RBC, BM samples from Ctnnb1^BM-CTL^ and Ctnnb1^BM-GOFwt/fl^ animals were isolated and cultured in MethoCult medium to perform a colony-forming unit (CFU) assay. Both groups were treated with 4-hydroxy tamoxifen (4-OHT) and colonies were microspically inspected and counted after 10 days of cultivation (Figure 6A–C). Most colony types (CFU-GEMM: colony-forming unit granulocyte, erythroid, macrophage, megakaryocyte; CFU-GM: colony-forming unit granulocyte, macrophage), -GM, -M, -EM, -G) showed a comparable level between the Ctnnb1^BM-CTL^ and Ctnnb1^BM-GOFwt/fl^ group. The CFU-MK and CFU-E, however, revealed a trend of increased megakaryocytic colonies while the number of erythroid colonies was decreased in the Ctnnb1^BM-GOFwt/fl^ condition (Figure 6B,C). Microscopic inspection of these colonies showed that especially in the Ctnnb1^BM-GOFwt/fl^ group, CFU-MKs were formed by highly matured individual MKs in contrast to those in the CTL group (Figure 6A). Finally, both groups had the same number of total, macroscopically visual colonies, confirming the comparability of both groups and suggesting that the MK and erythrocyte/RBC colonies were specifically affected. Recombination and pathway activation in vitro were corroborated on BM cells by genomic PCR and qRT-PCR for the Wnt/β-catenin target Axin2, respectively (Appendix A). As a positive control for Pdgfb-iCreERT2-mediated recombination and pathway activation, mouse brain microvascular endothelial cells (MBMECs) from the same mice were cultured and treated with 25 µM 4-OHT, resulting in Ctnnb1^lox(ex3)^ locus recombination and Axin2 target gene expression (Appendix A).

### 3.4. PDGFb-iCreERT2-Mediated Wnt/β-Catenin Pathway Activation in MKPs Drives Megakaryopoiesis by Regulating Key Transcriptions Factors

The observation of increased MKPs in the spleen and BM of Ctnnb1^BM-GOF^ animals (Figure 5C,D) raised the question of whether the activated Wnt/β-catenin signaling affected transcription factors, which were described to favor MK lineage differentiation such as RUNX1 and FLI1 [4]. Indeed, RUNX1 was significant, and FLI1 in trend increased in FACS-sorted MKPs of the spleen in Ctnnb1^BM-GOFwt/fl^ animals, supporting the interpretation that dominant activation of Wnt/β-catenin in MKPs may activate downstream effectors of MK differentiation (Figure 6D).

## 4. Discussion

The aim of this study was to investigate the effect of constitutive β-catenin activation in PDGFb-expressing cells in the hematopoietic system, focusing on megakaryocytes (MKs).

In the present study, we show that Ctnnb1^BM-GOF^ mutants developed a reproducible lethal phenotype with hyperkeratosis, a defect in hair growth, splenomegaly, lower erythrocyte counts going along with thrombocytosis, and increased MK formation 3 weeks after tamoxifen-mediated recombination (Figure 1). Furthermore, the animals were gasping for air suggesting anomalies in the pulmonary system although lung histology and inflammatory markers were unsuspicious. These phenotypic characteristics were independent of sex but were to some extent dose dependent, as the splenomegaly was at least in trend more pronounced in homozygous than in heterozygous mice (Figure 1D,E). Peripheral blood (PB) samples revealed an increased level of myeloid and a decreased level of lymphoid cells (Figure 1J). Specifically, MK cells were increased in the spleen and BM of Ctnnb1^BM-GOF^ mice, suggesting that activation of Wnt/β-catenin signaling in Pdgfb-expressing cells in the BM fosters megakaryopoiesis. Interestingly, thrombocytes that were at least in trend also increased in the Ctnnb1^BM-GOFwt/fl^ group, may suggest that the release of platelets was affected in these animals (Figure 1F).

The lethal phenotype, however, was not caused by changes in the hematopoietic system or in the endothelium (Appendix A). It should be noted that mice in which β-catenin is conditionally activated in ECs at postnatal stages develop fatal heart failure after 40 weeks post activation of the pathway [20]. Because of this, it is not likely that the endothelium was the main reason for the lethal Ctnnb1^BM-GOF^ mutants in this study since the phenotype already occurred in the third week after tamoxifen treatment. Along the line of possible vascular effects in the hematopoietic system, it is important to keep in mind that the vasculature of the murine BM shows considerable diversity, displaying arterial endothelial cells (AEC), H-type capillary vessels and bone marrow sinusoidal (L-type) ECs. The latter type contributes to osteogenesis, bone angiogenesis and HSPC maintenance and differentiation [21,22]. L-type, sinusoidal ECs have indeed been shown to control erythroid differentiation and reticulocyte maturation via Wnt/β-catenin signaling [10]. The findings by Heil and colleagues, however, were identified to be EC driven, which was not the case for the increase in MkPs in the present study, as suggested by the lack of effects in the Ctnnb1^EC-GOFwt/fl^ transgenic mice (Appendix A), as well as by the BMT of the Ctnnb1^BM-GOFwt/fl^ (Figure 5) and the in vitro CFU assay (Figure 6). To conclusively clarify the role of β-catenin signaling in the vasculature versus the MEP lineage in the BM, its physiological activation needs to be analyzed, and more specific Cre driver mouse lines for the MK lineage need to be established and used to activate or inhibit the pathway.

In the past, β-catenin signaling in the bone marrow has been attributed to several physiological and pathological conditions, ranging from maintenance of hematopoietic stem cells (HSCs), progression of drug-resistant mixed lineage leukemia (MLL) and formation of myeloproliferative neoplasms (MPNs) [23,24]. As initially mentioned, the genetic deletion of β-catenin caused reduced leukemia incidence and a lowering of stem cell self-renewal in AML mouse models [8].

Megakaryocytes (MKs) belong to the largest cells (50μm–100μm) in the BM. Together with HSCs, they are considered the rarest populations in the BM [25]. Their main function is to produce platelets via endomitosis, which are essential for homeostasis, wound healing, angiogenesis, inflammation and innate immunity. The main driving force for platelet formation is the regulator thrombopoietin (TPO), via its MK-specific receptor myeloproliferative 24 leukemia protein (c–Mpl, CD110), thereby fostering proplatelet, pre-platelet and ultimately thrombocyte formation [26,27].

To study the role of β-catenin in MKs and potentially also in endothelial cells, we generated double-transgenic mice resulting in Ctnnb1^BM-GOFwt/fl^ or Ctnnb1^BM-GOFfl/fl^ with single transgenic animals lacking the PDGFb-CreERT2 construct as controls (Ctnnb1^+/lox(ex3)^, CTL).

More specifically, we could show that in the BM the vast majority of all GFP^+^ cells were also positive for the megakaryocytic marker CD41 and the progenitor cell marker CD150 (Figure 3). Given that CD41 is a highly specific marker for MKs and platelets, whereas CD150 serves as a marker for HSC [28,29,30,31,32], the GFP^+^/CD41^+^/CD150^+^ population defines MK progenitors (MKPs) [33,34]. These findings brought us to the conclusion that β-catenin gain of function (GOF) is induced under the Pdgfb promoter at an early stage of megakaryopoiesis close to the branching point to MEPs. Although the efficient, Cre-mediated recombination in MKs by the Pdgfb-CreERT2 line has previously been shown, it would be interesting to further investigate in more detail which target genes are activated in MKPs by β-catenin to directly prove pathway activation [13].

Given that the Wnt/β-catenin pathway activation is known to promote the self-renewal and repopulation capacity of HSCs [11,12], we analyzed HSCs and progenitor cells, revealing that higher hierarchic hematopoietic cells (HSCs, GMPs, CLPs and MEPs) were not affected in BMT_Ctnnb1^BM-GOF^ (Figure 4; Appendix A). This in turn suggests that the applied animal model in this study is highly specific for megakaryocytic cells as discussed above.

Knowing that the PDGFb transgene is expressed in megakaryocytic cells [13], it was a logical step to test whether the lethal phenotype occurred because of changes in the hematopoietic system. Bone marrow transplantation (BMT) experiments revealed that lethality was not due to changes in hematopoiesis and megakaryopoiesis (Figure 5). However, FACS analysis on BM and spleen samples of these animals indicated that MKPs significantly increased in the BMT-Ctnnb1^BM-GOFwt/fl^ group, whereas the BMT-Ctnnb1^BM-GOFfl/fl^ showed only a trend towards increased MK counts (Figure 5).

Moreover, the same MKPs were expressing the transgene GFP reporter in recipients with mutant BM, proving the validity of the BMT model (Appendix A). Finally, CBC assays showed that BMT-Ctnnb1^BM-GOFwt/fl^ mutants developed mild anemia, likely due to relative megakaryocytic skewing, and largely compensated by splenic emergency erythropoiesis (Appendix A), which was reminiscent of the investigation in the parental transgenic animals. Hence, the BMT experiments support the hypothesis that activated Wnt/β-catenin signaling in Pdgfb-expressing cells, which we identified as MKPs, acts on cell-fate decision by favoring MK differentiation at the expense of erythropoiesis. That this effect was highly specific for MKs was highlighted by the finding that no alterations in the HSC, GMP and CLP pools between CTLs and Ctnnb1^BM-GOF^ were detectable (Appendix A). These data corroborated published data on the expression of Pdgfb in MKs [13].

The apparently stronger phenotype in heterozygous GOF mice might be related to more efficient downstream signaling of β-catenin if both, full-length and truncated proteins are present. A mechanistic explanation, however, requires further investigation.

The identification of a CD41^-^/GFP^+^ cell population in the BM from Ctnnb1^BM-GOF^ might be related to the recent finding that HSCs and multipotent progenitors 2 (MPP2) can directly differentiate into MKPs [35]. Possibly, the CD41^-^/GFP^+^ cells are MPP2 cells priming for an early stage of MKP differentiation by activating Pdgfb expression. This interpretation, however, requires further investigation, given that several other cell types express Pdgfb.

The first conclusion was that the Ctnnb1^BM-GOF^ specifically caused the boosting of megakaryopoiesis by forcing MEPs towards MK differentiation, thereby leading to the exhaustion of the MEP pool. Interestingly, this assumption could not be confirmed in the BMT experiments in which MEP levels were comparable between BMT CTLs and BMT Ctnnb1^BM-GOF^ (Appendix A). Consequently, the decrease in MEPs in parental mutants likely developed because of other reasons.

To understand if the augmented MK formation in β-catenin GOF mice is a cell-autonomous effect, it was necessary to exclude the influence of the BM stromal tissue on megakaryocytic progenitors in the context of activated Wnt/β-catenin signaling. Essentially, the CFU assays in vitro with BM samples from CTL and Ctnnb1^BM-GOF^ animals supported the interpretation that β-catenin signaling indeed drives MK differentiation in a cell-autonomous fashion (Figure 6). Nevertheless, further investigation is required to determine the exact timing as well as further narrowing down the cell-specificity of Wnt/β-catenin activation in erythrocyte and megakaryocyte differentiation.

Finally, downstream pathways involved in MK differentiation, involving transcription factors (TFs) such as friend leukemia integration 1 (FLI1), runt-related transcription factor (RUNX1) and GA binding protein alpha (GABPα), were reported to orchestrate and coordinate megakaryopoiesis. Herein, RUNX1 regulates polyploidization by silencing MYH10, which mediates the transition from mitosis to endomitosis. At later stages, it promotes an arrest of endomitosis by the upregulation of p19INK4D [36,37,38,39,40]. Furthermore, Kuvardina and colleagues described RUNX1 to inhibit erythroid differentiation by downregulating krueppel-like factor 1 (KLF1), thereby promoting megakaryocytic lineage differentiation [4]. The authors further describe that at the branching point of MEPs, enhanced RUNX1 influences the balance between KLF1 and FLI1 towards FLI1, supporting MK and preventing erythroid differentiation. Together with FLI1, Runx1 counteracts the activity of T-cell acute lymphocytic leukemia protein 1 (TAL1), resulting in the inhibition of erythropoiesis [4]. Moreover, it was previously shown in mice and humans that mutations in Nfe2, Fli1 and Runx1 cause thrombocytopenia, and that promoter occupancies with NFe2, Fli1 and Runx1 correlated with terminal MK differentiation [41].

Given the fact that Wnt/β-catenin signaling plays a major role in the expression of target genes, such as Runx2, we asked if the activation of β-catenin in hematopoietic progenitors can influence the expression of Runx1, consequently affecting megakaryopoiesis and erythropoiesis [42]. Indeed, in FACS-sorted MKPs, qRT-PCR revealed Runx1 to be significant and FLI1 in trend upregulated cells of Ctnnb1^BM-GOF^ samples (Figure 6D).

Although we do not provide evidence for the direct regulation of Runx1 or Fli1 by β-catenin on the promoter level, our data strongly support a crucial role of Wnt/β-catenin signaling herein.

So far, it has been shown that the activation of β-catenin appears to influence the function of tumor-initiating fusion proteins such as between Ewing sarcoma (EWS) and FLI1 (EWS/Fli1) [43]. As proposed by Kuvardina et al., another possible scenario could be that RUNX1 may act together with TAL1 to maintain low-level expression of KLF1, while KLF1 activates erythroid and represses megakaryocytic genes [4]. Upon megakaryocytic differentiation, RUNX1 increases and acts as a transcriptional repressor on the KLF1 promoter. Thus, the negative influence of RUNX1 on KLF1 expression shifts the balance from KLF1 towards FLI1. Consequently, the erythroid gene expression program is shut down and megakaryocytic genes are upregulated (Figure 6E). In line with this, RUNX1 expression was already described to be decreased during erythroid differentiation [4,44].

So far there is little known regarding the upstream regulation in MKPs. Given that aberrant activation of the JAK2/STAT pathway is linked to the formation of MPNs (Myeloproliferative Neoplasms), which in some cases coincide with high levels of β-catenin activity, a causal connection is tempting to speculate [6,45]. Indeed, it has been shown that JAK2 can regulate β-catenin in T cell leukemic Jurkat and erythroleukemia HEL cells in vitro [46]. Moreover, in chondrocytes, JAK2/STAT3 activation by leptin leads to an upregulation of the Wnt receptors frizzled 1 and 7 (Fzd1, Fzd7), rendering chondrocytes more susceptible to Wnt3a-mediated Wnt pathway activation [46]. How and if JAK2/STAT and Wnt/β-catenin cooperate during megakaryopoiesis as well as in the context of myeloproliferative neoplasms requires additional investigations. To better determine the transcriptional role of β-catenin in the context of megakaryocyte and erythrocyte differentiation, loss-of-function (LOF) experiments for β-catenin in a cell-specific manner could be informative. These experiments have already been performed in a broader cellular context in the hematopoietic system, however, without taking the dual function of β-catenin as a structural protein at adherens junctions and as a co-transcription factor in the nucleus into account [47]. In general, it is critical during the process of cell differentiation, as well as in which cells β-catenin is activated. This also explains that different experimental setups in vitro and in vivo may lead to opposing results, i.e., inhibition versus promotion of MK differentiation [11,48,49]. Therefore, although not an easy task, future investigations need to determine the level of β-catenin activation during MK lineage specification.

## 5. Conclusions

In the present study, we propose that Wnt/ß-catenin signaling acts at the megakaryocyte/erythrocyte branching point in MEPs, skewing myeloid differentiation towards megakaryopoiesis (Figure 6E). Our data support a scenario in which Wnt/β-catenin acts upstream of RUNX1, shifting it to the protein–protein interaction with FLI1, which favors MK lineage differentiation and inhibits erythroid production [4,50].

This finding may help to define new strategies for the treatment of thrombocytosis and erythrocytosis in the context of MPNs ET and PV [51,52]. High-risk ET and PV patients require cytoreductive therapy with hydroxyurea as the first line [53], and interferon-α and busulfan as second-line drugs [52,54,55]. However, the treatment lacks the confirmation of long-term safety and superiority, asking for novel molecular targets associated with ET and PV.

## Figures and Tables

**Figure 1 cells-12-02765-f001:**
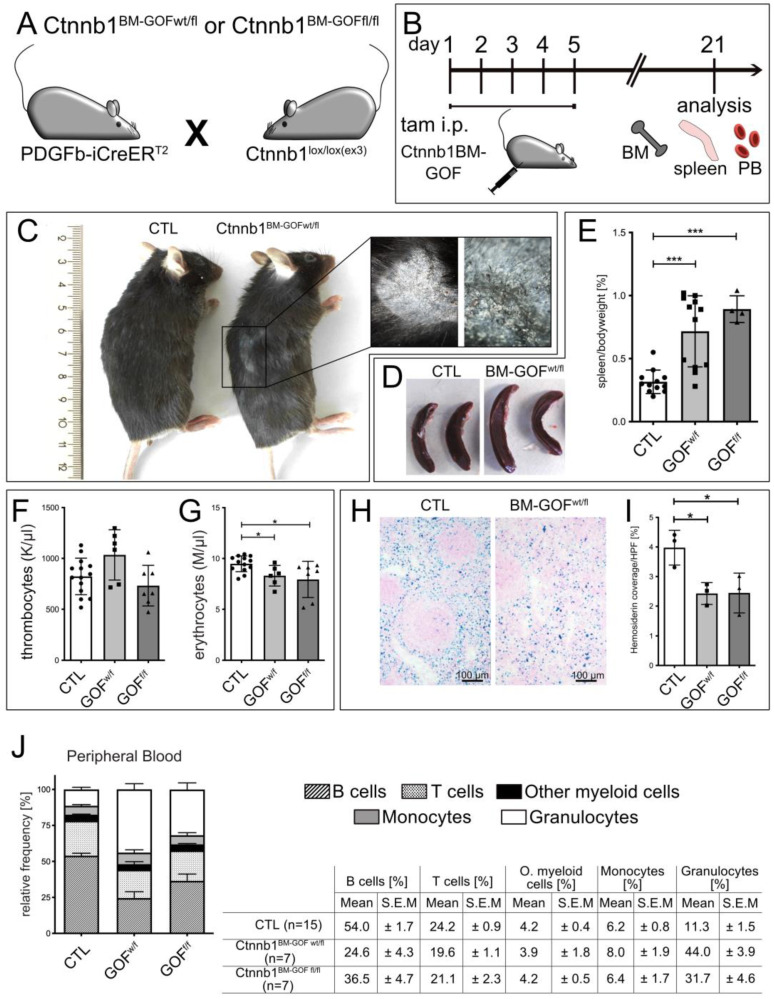
(**A**) Crossing scheme of the single transgenic mice PDGFb-iCreERT2 and Ctnnb1*^wt/lox(ex3)^* or Ctnnb1*^lox/lox(ex3)^* to obtain double transgenic hetero- (Ctnnb1^BM-GOFwt/fl^) and homozygous (Ctnnb1^BM-GOFfl/fl^) mice. (**B**) Schedule of tamoxifen injection of double transgenic mice to achieve constitutive activation of β-catenin in MKPs of the BM. Animals were sacrificed 21 days after the last tamoxifen injection. (**C**) Gross phenotype of mice at 21 days. Inset shows higher magnification of the flaked skin with hair loss in Ctnnb1^BM-GOFw/fl^ mice. (**D**) Gross morphological phenotype of spleen derived from Ctnnb1^BM-CTL^ and Ctnnb1^BM-GOFw/fl^ mice (*n* = 4). (**E**) Quantification of spleen to body weight of Ctnnb1^BM-CTL^ (*n* = 12), Ctnnb1^BM-GOFw/fl^ (*n* = 12) and Ctnnb1^BM-GOFfl/fl^ (*n* = 4) mice. (**F**) Complete blood cell (CBC) assay from peripheral blood (PB) showing thrombocytes (K/µL) for Ctnnb1^BM-CTL^ (*n* = 12), Ctnnb1^BM-GOFw/fl^ (*n* = 6) and Ctnnb1^BM-GOFfl/fl^ (*n* = 7) mice. (**G**) CBC assay from PB showing erythrocytes/RBCs (M/µL) for CTL (*n* = 12), Ctnnb1^BM-GOFw/fl^ (*n* = 6) and Ctnnb1^BM-GOFfl/fl^ (*n* = 7) mice. (**H**) Prussian blue staining of spleen cryosections from spleen of Ctnnb1^BM-CTL^ and Ctnnb1^BM-GOFw/fl^ mice and (**I**) quantification of hemosiderin coverage of Ctnnb1^BM-CTL^ (*n* = 3), Ctnnb1^BM-GOFw/fl^ (*n* = 3) and Ctnnb1^BM-GOFfl/fl^ (*n* = 3) mice using ImageJ software 2.9.0/1.53t. (**J**) Quantitative FACS analysis of leukocytes (myeloid cells: granulocytes, monocytes; lymphoid cells: B and T cells) in peripheral blood Ctnnb1^BM-CTL^ (*n* = 15), Ctnnb1^BM-GOFw/fl^ (*n* = 7) and Ctnnb1^BM-GOFfl/fl^ (*n* = 7). Significance is indicated as follows: *, *p* < 0.05; ***, *p* < 0.001.

**Figure 2 cells-12-02765-f002:**
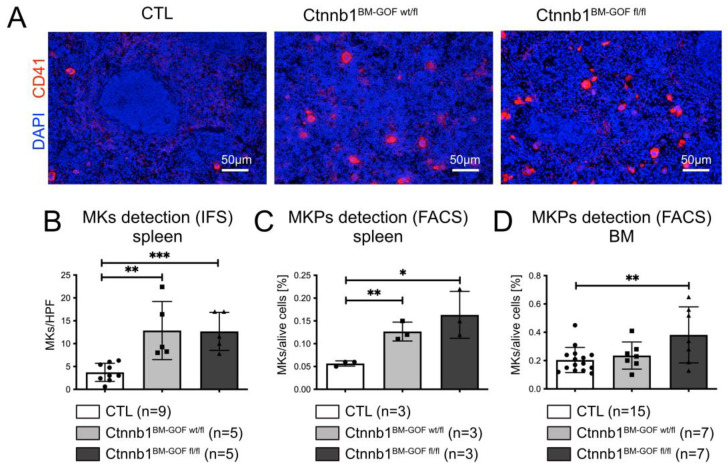
MK presence in spleen and BM. (**A**) Exemplary nuclear staining by DAPI (blue) and CD41 (red) in the spleen of Ctnnb1^BM-CTL^ (*n* = 9), Ctnnb1^BM-GOFw/fl^ (*n* = 9) and Ctnnb1^BM-GOFfl/fl^ (*n* = 4) mice. Scale Bar: 50μm. (**B**) Quantification via counting of CD41^+^ MKs cells per HPF on spleen cryosections of Ctnnb1^BM-CTL^ (*n* = 9), Ctnnb1^BM-GOFw/fl^ (*n* = 5) and Ctnnb1^BM-GOFfl/fl^ (*n* = 5) mice. (**C**) Spleen CD41^+^/CD150^+^ MKPs were quantified by FACS analysis of Ctnnb1^BM-CTL^ (*n* = 3), Ctnnb1^BM-GOFw/fl^ (*n* = 3) and Ctnnb1^BM-GOFfl/fl^ (*n* = 3) mice. (**D**) Bone marrow CD41^+^/CD150^+^ MKPs were quantified by FACS analysis of Ctnnb1^BM-CTL^ (*n* = 15), Ctnnb1^BM-GOFw/fl^ (*n* = 7) and Ctnnb1^BM-GOFfl/fl^ (*n* = 7) mice. Significance is indicated as follows: *, *p* < 0.05; **, *p* < 0.01; ***, *p* < 0.001.

**Figure 3 cells-12-02765-f003:**
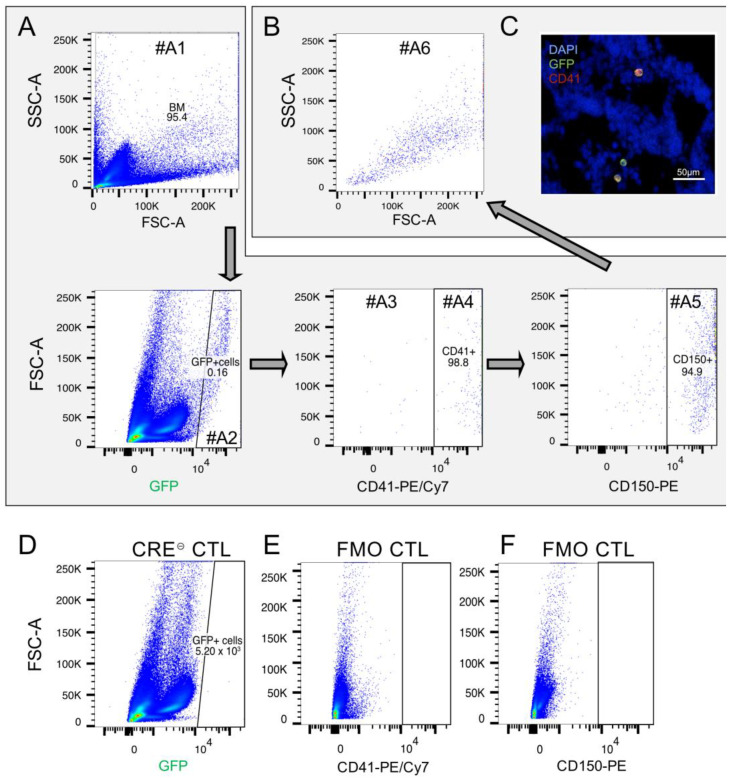
Analysis of the PDGFb transgene-expressing cells in the spleen. (**A**) To detect the expression pattern of the PDGFb construct in spleen cells, PDGFb transgene reporter (GFP) was detected using FACS analysis on spleen samples. GFP-positive cells (gate #A2) were mainly CD41^+^/CD150^+^ (gate #A4, #A5) showing (**B**) big cell volume and high granularity (gate #A6) indicating megakaryocytic lineage properties. There was the rest of unidentified GFP^+^/CD41^-^ cells, which were detected by FACS (gate #A3) as well as by (**C**) immunofluorescent staining on cryosections. Blue, DAPI; green, GFP; red, CD41. (**D**) Cre^⊝^ CTL spleen cells stained for GFP. (**E**) Fluorescence minus one (FMO) control for CD41-PE/Cy7. (**F**) Fluorescence minus one (FMO) control for CD150-PE.

**Figure 4 cells-12-02765-f004:**
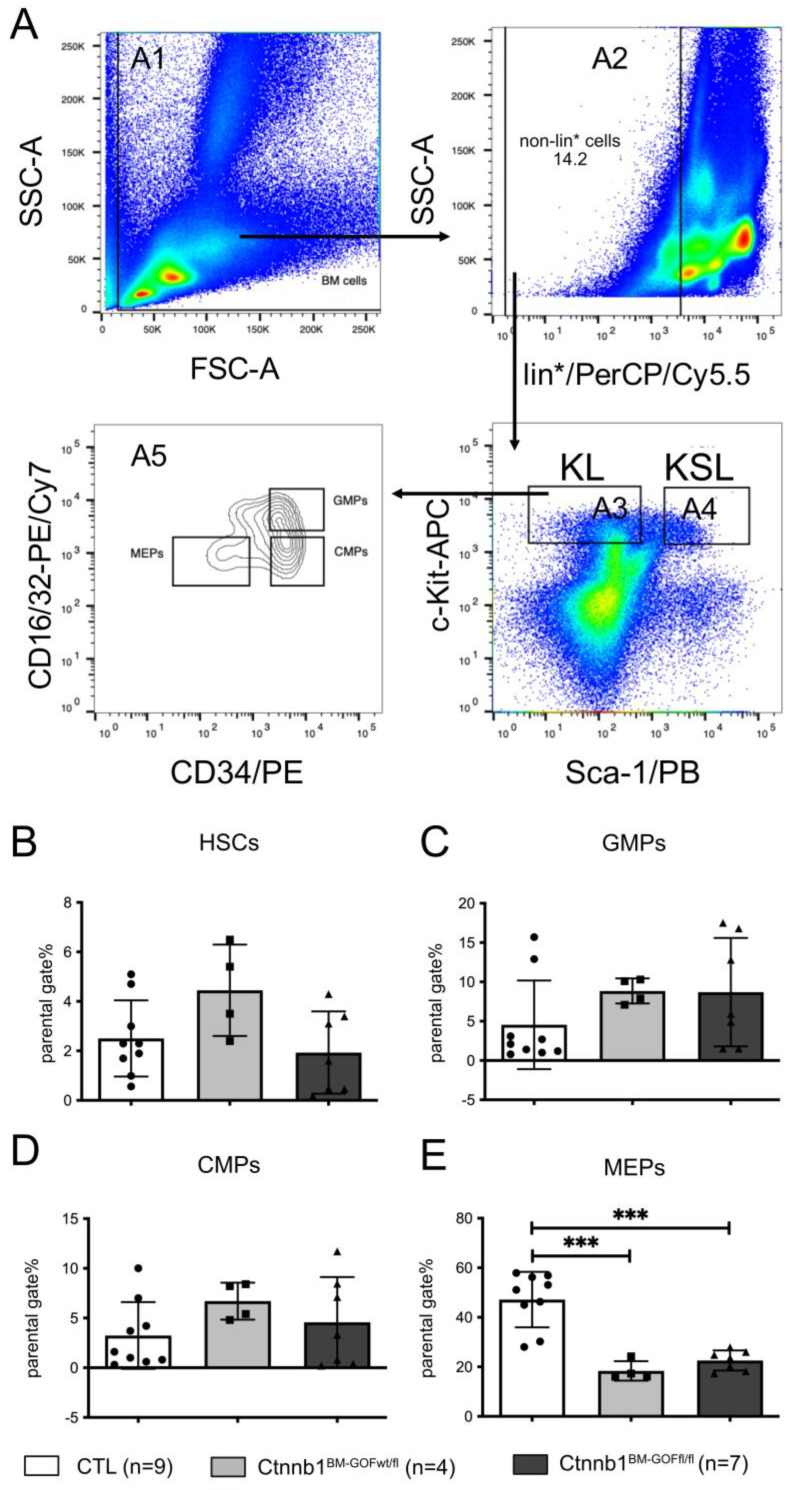
FACS analysis of hematopoietic stem cells (HSCs) and progenitors in the BM. (**A**) BM-gating strategy: KSL cells (A4: LT-HSCs, ST-HSCs, MPPs) and KL cells (A3: CMPs, GMPs, MEPs) were gated out of lin^-^ cells (A2); 1 × 10^6^ events were recorded using FACS ARIA. (**B**) HSCs, (**C**) granulocyte-monocyte (GMPs), (**D**) common myeloid (CMPs) and (**E**) megakaryocyte–erythrocyte (MEPs) progenitors in the BM of Ctnnb1^BM-CTL^ (*n* = 9), Ctnnb1^BM-GOFw/fl^ (*n* = 4) and Ctnnb1^BM-GOFfl/fl^ (*n* = 7) mice after tamoxifen treatment. Significance is indicated as follows: ***, *p* < 0.001.

**Figure 5 cells-12-02765-f005:**
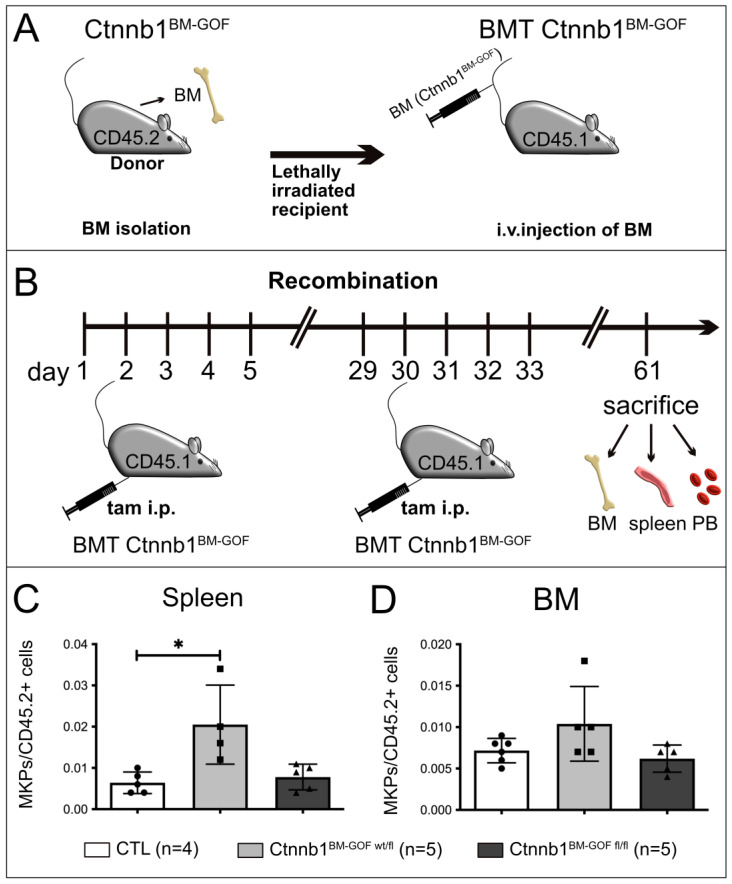
(**A**) Scheme of bone marrow transplantation (BMT) from CD45.2/Ctnnb1BM-GOF donor into lethally irradiated CD45.1/CTL recipient mice. (**B**) Schedule of tamoxifen (tam) injection and organ collection from successfully engrafted CD45.1/BMT Ctnnb1^BM-GOF^ mice. (**C**) FACS analysis of spleen MKPs from CTL (*n* = 5), Ctnnb1^BM-GOFw/fl^ (*n* = 4) and Ctnnb1^BM-GOFfl/fl^ (*n* = 5) mice after tamoxifen treatment. (**D**) FACS analysis of bone marrow (BM) MKPs from Ctnnb1^BM-CTL^ (*n* = 6), Ctnnb1^BM-GOFw/fl^ (*n* = 5) and Ctnnb1^BM-GOFfl/fl^ (*n* = 5) mice after tamoxifen treatment. Significance is indicated as follows: *, *p* < 0.05.

**Figure 6 cells-12-02765-f006:**
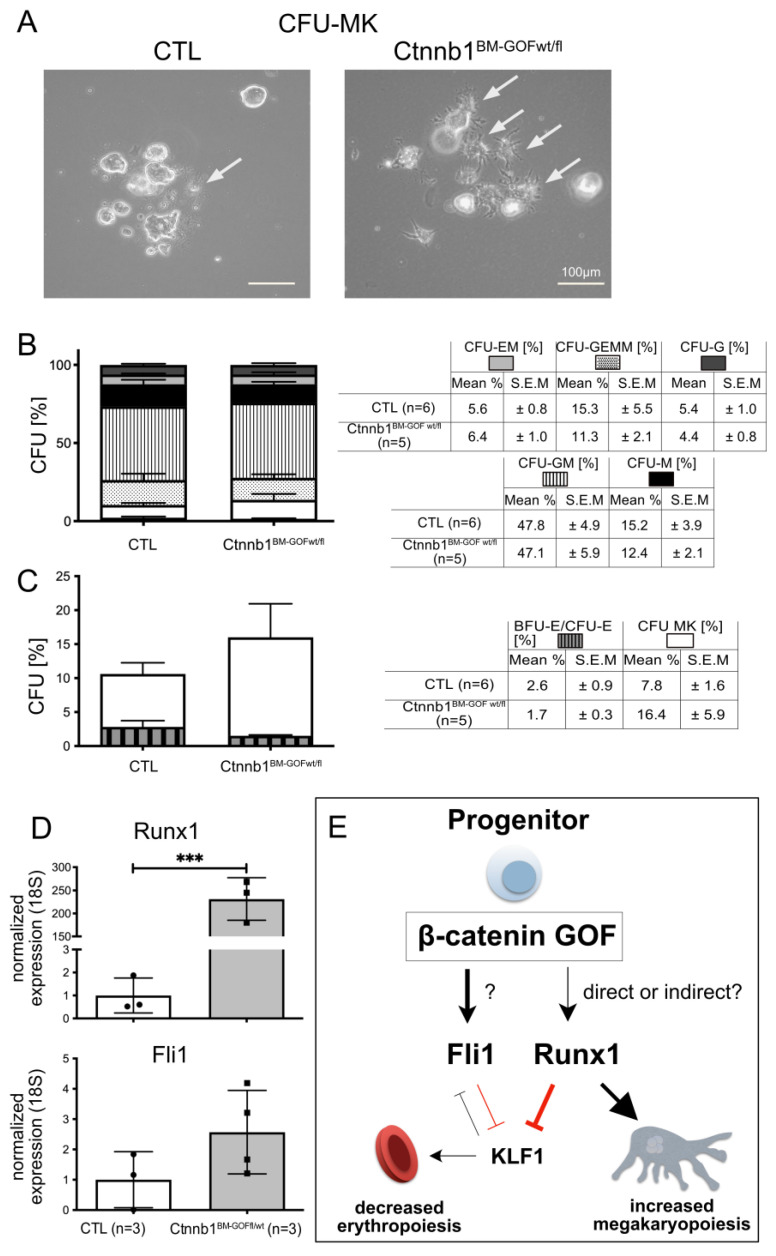
Colony forming unit assay (CFU) assay with BM samples of Ctnnb1^BM-GOF^ and Ctnnb1^BM-CTL^ animals. (**A**) Isolated BM nucleated cells cultured in MethoCult medium including TPO and 4-OHT after 10 days of cultivation. Arrows point to cells with MK morphology. (**B**) Quantification of CFU types in BM cells from CTL (*n* = 6) and Ctnnb1BM-GOF^wt/fl^ (*n* = 5) based upon microscopically visible colonies (BFU-E/CFU-E, CFU-MK, -GEMM, -GM, -M, -EM, -G). (**C**) Comparison of BFU-E/CFU-E and CFU-MK downstream of MEP in BM cells from CTL (*n* = 6) and Ctnnb1BM-GOF^wt/fl^ (*n* = 5) mice. Scale bar: 100 μm. (**D**) Analysis of MK-specific *Runx1* and *Fli1* transcription factors (TFs) expression in FACS sorted MKPs of the spleen from CTL (*n* = 3) and Ctnnb1_BM-GOFwt/fl_ (*n* = 3). All expression levels were normalized to 18s. (**E**) Model of how β-catenin GOF might favor megakaryopoiesis and repress erythropoiesis. Constitutively activated Wnt/β-catenin signaling favors megakaryocytic cell-fate decision by increasing FLI1 and RUNX1 expression, which reinforces the inhibitory effect on erythropoiesis via inhibiting krueppel-like factor 1 (KLF1). Significance is indicated as follows: ***, *p* < 0.001.

**Table 1 cells-12-02765-t001:** Antibodies.

Antibody	Host	Company	Catalog #	Dilution	Method
ABs for Immunofluorescence and Immunohistochemistry					
GFP	Chicken	Abcam (Cambridge, UK)	ab13970	1:500	PFA
CD41	Rat	BD Biosciences (Heidelberg, Germany)	553847	1:200	PFA
Ki67 (SP6)	Rabbit	ThermoFisher Scientific (Darmstadt, Germany)	RM-9106S0	1:200	PFA
Donkey α Rat 550	Donkey	ThermoFisher Scientific (Darmstadt, Germany)	SA5-10027	1:200	PFA
Goat α Chicken 488	Goat	Abcam	Ab150169	1:200	PFA
ABs for flow cytometry (FACS)					
CD3-PE/Cy7		BioLegend (Amsterdam, The Netherlands)	100220	1:70	
CD4- PerCP		BD Biosciences (Heidelberg, Germany)	553052	1:50	
CD11b-v450		BD Biosciences (Heidelberg, Germany)	560456	1:50	
CD16/32-PE/Cy7		BioLegend	101318	1:75	
CD19-PE		BD Biosciences (Heidelberg, Germany)	557399	1:50	
CD34-PE		BD Biosciences (Heidelberg, Germany)	551387	1:75	
CD41- PE/Cy7		BioLegend (Amsterdam The Netherlands)	133916	1:50	
CD45.1-PerCP/Cy5.5		BioLegend (Amsterdam The Netherlands)	110727	1:70	
CD45.2-BV510		BioLegend (Amsterdam The Netherlands)	109837	1:70	
CD150-PE		BioLegend (Amsterdam The Netherlands)	115904	1:50	
c-Kit-APC		BD Biosciences (Heidelberg, Germany)	553356	1:75	
F4/80-APC		eBioscience (Darmstadt, Germany)	17-4801-82	1:50	
GFP-Alexa 488		Life Technologies (Darmstadt, Germany)	A-21311	1:400	PFA
Lineage Panel: (TER-119, CD11b, Ly-6G/Ly-6C, CD3e, CD45R/B220)-Biotin		BioLegend (Amsterdam The Netherlands)	133307	1:300	
Ly6G-FITC		BD Biosciences (Heidelberg, Germany)	551460	1:50	
Sca-1-PB		BioLegend (Amsterdam The Netherlands)	122520	1:75	
Streptavidin-PerCP/Cy5.5		BioLegend (Amsterdam The Netherlands)	405214	1:300	

**Table 2 cells-12-02765-t002:** Primer sequences used for cDNA amplification by qRT-PCR.

Primer For	Sequence 5′-3′ Sense	Sequence 5′-3′ Antisense
RUNX1b	CCTCCGGTAGTAATAAAGGCTTC	CCGATTGAGTAAGGACCCTGAA
FLI1	CCCTGCAGCCACATCCAACAG	GGAGGATGGGTGAGACGGGAC
Rplp0	CTTTGGTCGCTCGCTCCTC	CTGACCGGGTTGGTTTTGAT
18s	GTGTTTGACAACGGCAGCATT	TCTCCACAGACAATGCCAGGA

## Data Availability

The data presented in this study are available in the article and Appendix A.

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
