# Peer review of "Wnt/β-Catenin-Signaling Modulates Megakaryopoiesis at the Megakaryocyte-Erythrocyte Progenitor Stage in the Hematopoietic System"

_cells, 2023, doi:10.3390/cells12232765_

Round 1
Reviewer 1 Report
Comments and Suggestions for Authors
In this manuscript, Yalcin et al., have demonstrated the role of Wnt/beta-catenin signaling axis in biasing the differentiation process towards the megakaryocyte lineage from the megakaryocyte-erythrocyte stage by making use of mice expressing beta-catenin with gain of mutation from hematopoietic cells expressing, platelet-derived-growth-factor-beta.
Using multiple experimental approaches, they show that beta-catenin with gain of function mutation, preferentially enables the differentiation of megakaryocyte-erythrocyte-progenitors towards the megakaryocyte lineage and suppresses erythropoiesis by activating and inhibiting Runx1 and Fli1 transcription factors respectively. This study greatly enhances our understanding of megakaryopoiesis from the megakaryocyte-erythrocyte progenitors. Experiments were well designed with appropriate controls. There are some concerns which the authors need to address to ensure publication of the manuscript.
Major concerns:
How did the authors ensure the Wnt pathway was indeed upregulated in megakaryocyte-erythrocyte progenitors expressing beta-catenin with the gain of function mutation? This needs to be confirmed.
Minor concerns:
While the authors have used qRT-PCR to demonstrate the switch in transcription factors in megakaryocyte-erythrocyte progenitor cells expressing beta-catenin with gain of function mutations, further experiments with cells lacking beta-catenin or beta-catenin with loss of function are required to confirm the role of Wnt signaling in skewing the differentiation process.
Comments on the Quality of English LanguageQuality of English language is good
Author Response
Point-by-point response:
Reviewer #1:
Comments and Suggestions for Authors
In this manuscript, Yalcin et al., have demonstrated the role of Wnt/beta-catenin signaling axis in biasing the differentiation process towards the megakaryocyte lineage from the megakaryocyte-erythrocyte stage by making use of mice expressing beta-catenin with gain of mutation from hematopoietic cells expressing, platelet-derived-growth-factor-beta.
Using multiple experimental approaches, they show that beta-catenin with gain of function mutation, preferentially enables the differentiation of megakaryocyte-erythrocyte-progenitors towards the megakaryocyte lineage and suppresses erythropoiesis by activating and inhibiting Runx1 and Fli1 transcription factors respectively. This study greatly enhances our understanding of megakaryopoiesis from the megakaryocyte-erythrocyte progenitors. Experiments were well designed with appropriate controls. There are some concerns which the authors need to address to ensure publication of the manuscript.
Major concerns:
How did the authors ensure the Wnt pathway was indeed upregulated in megakaryocyte-erythrocyte progenitors expressing beta-catenin with the gain of function mutation? This needs to be confirmed.
Authors response: we thank the reviewer for this critical comment. Given that the generation of a Ctnnb1BM-GOF mouse requires a new animal experiment that on its own requires already 3 weeks after tamoxifen induction (Fig. 1B), we were not able to perform a dedicated experiment within the 10days of provided time for revision. However, we have performed a qRT-PCR analysis on cDNA of FACS-sorted MEPs from spleen, of which we have had some residual sample frozen. Unfortunately, the remaining cDNA was too little to give robust results for Axin2 induction, which is a well-characterized target gene of β-catenin transcriptional activation.
Due to the low number of MKs in the bone marrow, we did not generate samples for western blot, which would, due to its low sensitivity, require a high number of sorted MKs. Consequently, we cannot provide these data at this point.
Nevertheless, we have shown in Fig.3 that the majority of GFP+ cells, which do express the Cre recombinase, are indeed CD41+/CD150+ MKPs. Although indirect, this finding highly suggest that recombination can only take place in GFP+ cells, hence in the MKP population, supporting the original publication on the Pdgfb-CreERT2 mouse line.
However, we are fully aware that direct proof of pathway activation would be ideal and therefore, we have added a sentence in the discussion to clarify this point. Lines 528-31
Minor concerns:
While the authors have used qRT-PCR to demonstrate the switch in transcription factors in megakaryocyte-erythrocyte progenitor cells expressing beta-catenin with gain of function mutations, further experiments with cells lacking beta-catenin or beta-catenin with loss of function are required to confirm the role of Wnt signaling in skewing the differentiation process.
Authors response: we thank the reviewer for this important comment. Indeed, β-catenin deletion specifically in the myeloid lineage or even in MEPs could be an interesting experiment that might contribute valuable aspects to the general understanding of the Wnt/β-catenin pathway in lineage determination in the hematopoietic system. Although β-catenin deletion in the hematopoietic system has already been performed, many questions remain unanswered. This is, among other reasosn, related to the dual function of β-catenin as a structural protein at cell-cell junctions and as a co-transcription factor. Hence, interpreting the effect of a knock-out (loss-of-function) is much more difficult than interpreting the effects of the gain-of-function condition. The ideal deletion approach would require maintaining the structural function of β-catenin, while deleting the transcriptional function. Although possible, these experiments are way beyond the scope of the present manuscript. We now have added a sentence to the discussion to raise this point. Lines 621-32
Reviewer 2 Report
Comments and Suggestions for Authors
General suggestions
Introduction
- Line 75: What the abbreviations PMF, PV and ET stand for should be written out.
Materials and Methods
- Line 98-112: Is this part of the writing requirement? Why is it still in the manuscript? What's the point?
- The data analysis method needs to be explained. How to carry out the difference analysis?
Reults
- Figure 1 J: Why are the proportions of different types of cells in Figure 1 J different from those in the table? With a different number of samples?
Comments on the Quality of English LanguageMinor editing of English language required
Author Response
Point-by-point response:
Reviewer #2:
Comments and Suggestions for Authors
General suggestions
Introduction
-Line 75: What the abbreviations PMF, PV and ET stand for should be written out.
Authors response: we thank the reviewer for this remark. The abbreviations are now explained in the text Lines 75-76, when first mentioned.
Materials and Methods
-Line 98-112: Is this part of the writing requirement? Why is it still in the manuscript? What's the point?
Authors response: we thank the reviewer for raising this point. This section has been removed.
-The data analysis method needs to be explained. How to carry out the difference analysis?
Authors response: we thank the reviewer for raising this critical point. We have now explained in more detail in particular the qRT-PCR (Lines 220-34) and added a new section on the statistical analysis as well as on Graphics and artwork Lines 281-89.
Results
-Figure 1 J: Why are the proportions of different types of cells in Figure 1 J different from those in the table? With a different number of samples?
Authors response: we thank the reviewer for raising this important point. The graph was correct but by accident the table was derived from the spleen samples instead of the peripheral blood samples. We have now added the correct table and have adapted the Figure 1J and the legend accordingly.
Reviewer 3 Report
Comments and Suggestions for Authors
Using two approaches, transgenic mouse models and bone marrow transplantation experiments, the authors show that a hyperactivated form of beta-catenin, expressed under the PDFG promoter, promotes megakaryopoiesis at the expenses of erythropoiesis.
The experiments are well designed and performed and the results are potentially interesting. However, there are some issues that need to be addressed.
Major points.
There is a lack of consistency in some of the results obtained by the two approaches, raising questions about the robustness and reliability of the results. Even more so considering that other authors have published a negative role of b-catenin in megakaryopoiesis (e.g., Sardina et al., BBA – Mol. Cell Res. 1843:2886, 2014; Paluru et al., Stem Cell Res. 12:441, 2014).
Specific comments:
1) There is no evidence of b-catenin hiperactivation. A WB must be submitted demonstrating that the truncated form of beta-catenin is expressed in transgenic MKs. It would also be desirable to show the phosphorylation status of this isoform.
2) The putative MK population in Figure 3B (#6) is very dispersed and not all cells have high SSC-A and FSC-A. Explain.
3) Very dispersed populations are also seen in #4 and #5, with events too close to the margins.
4) Figure 4 is identical to Supp. Fig. 3. A different Figure 4 should be displayed.
5) Some experiments where the differences appear to be significant (e.g., Supplementary Figure 9E) are not, and the opposite is true (e.g. Fig 1G). Data points should be displayed to show data dispersion.
6) Why an increase in megakaryopoiesis is observed in heterozygous but not homozygous mice (Fig. 5C). In general, the phenotype of the wt/f mice is stronger than that of f/f mice. This must be explained.
7) Results regarding Suppl. Fig. 11 and 12 should be explained properly in the Result Section, not in the Discussion. Furthermore, no specific information on the Cdh5PAC-Cre mouse model is provided in the corresponding figure legends.
8) Other authors have a described a negative role of beta-catenin in megakaryopoiesis. This needs to be discussed.
Minor points.
Does the lethal phenotype affect 100% of both heterozygous and homozygous animals? Any explanation for this lethality?
Numbers must be separated from units.
Figures should be mentioned in the text in numerical order
Are PDGFB-iCreERT2 mice homozygous for the Cre alele? This needs to be clarified.
Serious problems with labeling of figures. There is no homogeneity in the size of the letters. Some numbers and letters are too small.
Line 75: PMF, PV and ET should be defined here, not on line 95.
Line. 79: Change MG for MK.
Lines 84-86: Rephrase.
Lines: 98-112: Delete the first paragraph of M&M.
Line 118: PDGFB-iCreERT2 is repeated, as well as the adjacent reference, which is not numerical.
Line 128: Correct “promoter”.
Line 148: What “Tris ad” is? Tris concentration must be indicated.
Line 168: Leica is repeated.
Line 187: The composition of the FACS buffer must be indicated.
Line 203: The composition of the red blood lysis buffer must be indicated.
Lines 231-242. Text with different size letters and redundant parts.
In Figurentra 2B-D and 4B-E, the pattern of the bars does not correspond to the patterns of the labels. Suggestion: use color bars.
Figure 6A is mentioned after Figures 6B-D. Change the position of the panels.
Lines 518-520: This explanation should be in M&M.
Line: 623: Correct meg/ery
Comments on the Quality of English LanguageSome redundancies and spelling errors but not serious.
Author Response
Point-by-point response:
Reviewer #2:
Comments and Suggestions for Authors
General suggestions
Introduction
-Line 75: What the abbreviations PMF, PV and ET stand for should be written out.
Authors response: we thank the reviewer for this remark. The abbreviations are now explained in the text Lines 75-76, when first mentioned.
Materials and Methods
-Line 98-112: Is this part of the writing requirement? Why is it still in the manuscript? What's the point?
Authors response: we thank the reviewer for raising this point. This section has been removed.
-The data analysis method needs to be explained. How to carry out the difference analysis?
Authors response: we thank the reviewer for raising this critical point. We have now explained in more detail in particular the qRT-PCR (Lines 220-34) and added a new section on the statistical analysis as well as on Graphics and artwork Lines 281-89.
Results
-Figure 1 J: Why are the proportions of different types of cells in Figure 1 J different from those in the table? With a different number of samples?
Authors response: we thank the reviewer for raising this important point. The graph was correct but by accident the table was derived from the spleen samples instead of the peripheral blood samples. We have now added the correct table and have adapted the Figure 1J and the legend accordingly.
Reviewer #3:
Comments and Suggestions for Authors
Using two approaches, transgenic mouse models and bone marrow transplantation experiments, the authors show that a hyperactivated form of beta-catenin, expressed under the PDFG promoter, promotes megakaryopoiesis at the expenses of erythropoiesis.
The experiments are well designed and performed and the results are potentially interesting. However, there are some issues that need to be addressed.
Major points.
There is a lack of consistency in some of the results obtained by the two approaches, raising questions about the robustness and reliability of the results. Even more so considering that other authors have published a negative role of b-catenin in megakaryopoiesis (e.g., Sardina et al., BBA – Mol. Cell Res. 1843:2886, 2014; Paluru et al., Stem Cell Res. 12:441, 2014).
Authors response: we thank the reviewer for raising these critical points. We have now cited and discussed the references mentioned by the reviewer. In general, it is indeed still a matter of discussion what the detailed function of Wnt/β-catenin signaling in the hematopoietic system is. We have had raised this point in the Introduction “Herein, conflicting data regarding the outcome of Wnt/β-catenin activation in the hematopoietic system have challenged the understanding of its function so far. Depending on the use of cell-specific Cre-driver lines in mouse models, or on the cellular system used for in vitro differentiation assays have revealed that MKs and pro-platelet cells can either be increased or decreased [11,12].” And these discrepancies were part of the motivation to conduct the present study.
The major point we intend to make in this manuscript is that the timing, and cell-specific activation of the Wnt/β-catenin pathway is key to the understanding of its detailed role. This point is now explicitly raised in the Discussion. Lines 576-78.
Moreover, as also brought up by Reviewer #1, we have now discussed the options and limitations of β-catenin deletion approaches, taking into account that this protein has a dual function as a structural protein, and as a co-transcription factor. Lines 621-32.
Specific comments:
1) There is no evidence of b-catenin hyperactivation. A WB must be submitted demonstrating that the truncated form of beta-catenin is expressed in transgenic MKs. It would also be desirable to show the phosphorylation status of this isoform.
Authors response: we thank the reviewer for this critical comment. Given that the generation of a Ctnnb1BM-GOF mouse requires a new animal experiment that on its own requires already 3 weeks after tamoxifen induction (Fig. 1B), we were not able to perform a dedicated experiment within the 10days of provided time for revision. However, we have performed a qRT-PCR analysis on cDNA of FACS-sorted MEPs, which were used to demonstrate Runx1 regulation (Fig. 6D). Unfortunately, the remaining cDNA was too little to give robust results for Axin2 induction, which is a well-characterized target gene of β-catenin transcriptional activation.
Due to the low number of MKs in the bone marrow, we did not generate samples for western blot, which would, due to its low sensitivity, require a high number of sorted MKs. Consequently, we cannot provide these data at this point.
Nevertheless, we have shown in Fig.3 that the majority of GFP+ cells, which do express the Cre recombinase, are indeed CD41+/CD150+ MKPs. Although indirect, this finding highly suggest that recombination can only take place in GFP+ cells, hence in the MKP population, supporting the original publication on the Pdgfb-CreERT2 mouse line.
However, we are fully aware that direct proof of pathway activation would be ideal and therefore, we have added a sentence in the discussion to clarify this point. Lines 576-78.
2) The putative MK population in Figure 3B (#6) is very dispersed and not all cells have high SSC-A and FSC-A. Explain.
3) Very dispersed populations are also seen in #4 and #5, with events too close to the margins.
Authors response: we thank the reviewer for raising these critical points. The relatively broad distribution according to SSC-A and FSC-A in Figure 3B #A6 is the typical appearance of megakaryocytes of different maturity stages. Similarly, the CD41+ (Figure 3A #A4) and CD150+ (Figure 3A #A5) cells also show a relatively broad range of size (FSC-A).
4) Figure 4 is identical to Supp. Fig. 3. A different Figure 4 should be displayed.
Authors response: we thank the reviewer for raising this critical point. Suppl. Fig. 3 has been removed and the text has been adapted accordingly.
5) Some experiments where the differences appear to be significant (e.g., Supplementary Figure 9E) are not, and the opposite is true (e.g. Fig 1G). Data points should be displayed to show data dispersion.
Authors response: we thank the reviewer for raising this critical point. The figures mentioned have been adapted accordingly, as well as all figures for which individual data points provided additional information, without diminishing clarity (Stacked columns are still shown without individual data points).
6) Why an increase in megakaryopoiesis is observed in heterozygous but not homozygous mice (Fig. 5C). In general, the phenotype of the wt/f mice is stronger than that of f/f mice. This must be explained.
Authors response: we thank the reviewer for raising this interesting point. Although we noticed a slightly stronger effect of β-catenin GOF in the heterozygous compared to homozygous mice, we cannot explain this effect mechanistically. However, this has been mentioned by collaborators in personal communication even for other cellular systems. There is no published data on this effect, but our current hypothesis is that the transcriptional activation by truncated and therefore stabilized β-catenin, at best requires a pool of full length β-catenin that may sequester other binding partners such as classical cadherins and proteins of the degradation complex such as GSK3β. Although none of these speculations would fit into the present manuscript, we have added a sentence in the Discussion, raising the point. Lines 556-58.
7) Results regarding Suppl. Fig. 11 and 12 should be explained properly in the Result Section, not in the Discussion. Furthermore, no specific information on the Cdh5PAC-Cre mouse model is provided in the corresponding figure legends.
Authors response: we thank the reviewer for this comment. We have now described this approach first in the Results section. Lines 427-31.
Additionally, we have added an experimental outline for the endothelial GOF approach as Suppl. Fig. 10A.
8) Other authors have a described a negative role of beta-catenin in megakaryopoiesis. This needs to be discussed.
Authors response: we thank the reviewer for bringing up this important point. Indeed, this fact was, among other, the motivation to specifically investigate MK differentiation using the Pdgfb-CreERT2 line. The general problem is that activation of β-catenin signaling is highly cell type-, timing- and context-specific. Certainly, a detailed analysis of β-catenin activation in the bone marrow with regard to the different lineages would be an important study. We now raised this point in the Discussion. Lines 621-32.
Minor points.
Does the lethal phenotype affect 100% of both heterozygous and homozygous animals? Any explanation for this lethality?
Authors response: we thank the reviewer for asking this question. Yes, 100% of the heterozygouse and homozygouse Ctnnb1BM-GOF mice were lethal. We have analyzed the skin (Fig. 1C; Suppl. Fig 6, now Suppl. Fig. 5), as well as the lungs (Suppl. Fig 7, now Suppl. Fig. 6), although the skin unlikely is causal for the lethality. Unfortunately, we could not determine a clear reason for the lethal phenotype. It should be noted that according to the animal welfare regulations, we have had to sacrifice the mice due to their suffering. Therefore, it is not certain for all mice that they would have died within the same time frame.
Numbers must be separated from units.
Authors response: we thank the reviewer for pointing this out. We now have consistently separated numbers from units.
Figures should be mentioned in the text in numerical order
Authors response: we thank the reviewer for this comment. The figures now appear in the text in numerical order.
Are PDGFB-iCreERT2 mice homozygous for the Cre allele? This needs to be clarified.
Authors response: we thank the reviewer for this critical remark. We now clarified this in the “Material and Methods” section Lines 113-14.
Serious problems with labeling of figures. There is no homogeneity in the size of the letters. Some numbers and letters are too small.
Authors response: we thank the reviewer for this comment. We now have used same sized numbers and font in the different figures. Moreover, we have increased small sized numbers and letters for better readability.
Line 75: PMF, PV and ET should be defined here, not on line 95.
Authors response: we thank the reviewer for this remark. The abbreviations are now explained in the text Lines 75-76, when first mentioned.
Line. 79: Change MG for MK.
Authors response: we thank the reviewer for this remark. The abbreviations “MG” has now been changed to “MK” Line 80.
Lines 84-86: Rephrase.
Authors response: we thank the reviewer for this remark. The section has now been rephrased as follows: „Depending on the use of cell-specific Cre-driver lines in mouse models, or in cellular systems, conflicting data regarding the outcome of Wnt/β-catenin activation have challenged the understanding of its function in the hematopoietic system [11,12].“ Lines 83-86.
Lines: 98-112: Delete the first paragraph of M&M.
Authors response: we thank the reviewer for raising this point. This section has been removed.
Line 118: PDGFB-iCreERT2 is repeated, as well as the adjacent reference, which is not numerical.
Authors response: we thank the reviewer for this comment. The repetition of PDGFB-iCreERT2 as well as the wrong citation has been removed.
Line 128: Correct “promoter”.
Authors response: we thank the reviewer for this comment. The word “promotor” has now consistently been changed to “promoter”.
Line 148: What “Tris ad” is? Tris concentration must be indicated.
Authors response: we thank the reviewer for this comment. The information has been corrected to: “Bones (femur) were post fixed in 4 % PFA o/n at 4 °C. The decalcification process was performed in an EDTA-solution (10 % EDTA/10 mM Tris-HCL, pH 6.8-7.0) at 37 °C.”. Line 135.
Line 168: Leica is repeated.
Authors response: we thank the reviewer for this comment. The second “Leica” has been removed.
Line 187: The composition of the FACS buffer must be indicated.
Authors response: we thank the reviewer for this comment. The composition of the FACS buffer has now been indicated as follows: “After washing in FACS buffer (PBS + 5 % fetal calf serum, FCS), primary antibodies (Table 1) were applied for 1h.” Lines 176-77.
Line 203: The composition of the red blood lysis buffer must be indicated.
Authors response: we thank the reviewer for this comment. The composition of the red blood cell lysis buffer is not described in detail as it was purchased from Sigma-Aldrich/Merck. The description has now been changed as follows: “After centrifugation (1200 rpm, 4 °C), the pellet was re-suspended in 1ml red blood cell lysis buffer (Sigma-Aldrich/Merck, #11814389001) for 10 mins at RT, centrifuged and washed with FACS buffer.” Line 194.
Lines 231-242. Text with different size letters and redundant parts.
Authors response: we thank the reviewer for raising this point. The section on “RNA isolation and RT-PCR and qRT-PCR analyses” has now been changed accordingly. Line220-34.
In Figure ntra 2B-D and 4B-E, the pattern of the bars does not correspond to the patterns of the labels. Suggestion: use color bars.
Authors response: we thank the reviewer for this critical comment. The Figures have been modified for better clarity and readability.
Figure 6A is mentioned after Figures 6B-D. Change the position of the panels.
Authors response: we thank the reviewer for this remark. The text has been changed to fit the appearance of the figures. Line 443.
Lines 518-520: This explanation should be in M&M.
Authors response: we thank the reviewer for this remark. The text has been changed to fit the appearance of the figures. Line 117-19.
Line: 623: Correct meg/ery
Authors response: we thank the reviewer for this remark. The text has been changed accordingly. Lines 639-40.
Round 2
Reviewer 1 Report
Comments and Suggestions for Authors
The authors have tried their best address the concerns in the limited time that they had at their disposal.
However, the western and qPCR experiments that they attempted with limited samples will significantly improve the quality of the manuscript.
Comments on the Quality of English LanguageQuality of english is good.
Author Response
Authors response to both reviewers:
We thank the reviewers for the critical comment. We completely agree that the control of transgene recombination, proofing functionality of the model system, as well as confirmation of pathway activation by the recombined transgene is essential. We also thank Reviewer #2 for proposing more time for revision as well as the in vitro approach.
For the second revision, time was again limited and therefore, in vivo experiments were not possible within this timeframe. Consequently, and as suggested by Reviewer #2, we have cultivated bone marrow cells from PDGFb-iCreERT2wt/wt:Ctnnb1wt/lox(Ex3) (Ctrl) PDGFb-iCreERT2wt/cre:Ctnnb1wt/lox(Ex3) (GOF) mice with stem cell factor (100 ng/ml) and thrombopoietin (TPO, 100 ng/ml), and treated them every second day in culture with 25 µM 4-hydroxy-tamoxifen (4-OHT) and harvested the cells after 7 days.
The expectation was that only a small fraction of the cultured cells would differentiate into Pdgfb-expressing MKPs/MKs, we therefore isolated in parallel brain microvascular endothelial cells (MBMECs) from the same mice, which all express Pdgfb at high level. MBMECs were also cultured for 7 days. From both cells types gDNA and total RNA was isolated and recombination of the transgene and induction of the Wnt/β-catenin target Axin2 was analyzed by PCR and qRT-PCR, respectively. The data confirming recombination as well as Axin2 induction in GOF cells are now added to the manuscript as supplementary figure 12. We decided to add the novel data as supplementary figure, as the experiment could only be performed once (n=1).
Still we believe that this experiment and the analysis shows as proof-of-principle that the model system works, supporting the specificity of the acquired data.
Reviewer 3 Report
Comments and Suggestions for Authors
Although the manuscript has been substantially improved, there is still a fundamental question that has not been satisfactorily answered.
Showing the truncated beta-catenin protein is an essential control. Authors should request time from the editor to conduct this experiment. Three weeks is not a long time. Alternatively, in case it is not possible to obtain a sufficient amount of protein, a semiquantitative PCR should be performed to show the presence of the deleted fragment in the wt and its absence in the mutant. 7 days of bone marrow culture in the presence of TPO gives enough MK to perform this experiment.
Comments on the Quality of English LanguageThere are still some phrases that need to be revised, e.g. lines 623-624
Author Response

(The authors gave the same response as above.)

Round 3
Reviewer 1 Report
Comments and Suggestions for Authors
The authors have responded to my concerns and the manuscript is ready for publication.
Reviewer 3 Report
Comments and Suggestions for Authors
The authors have sufficiently addressed the majority of the concerns and the manuscript has notably improved. Based on my criteria, the article is now suitable for publication.